# Factors Associated with Dog Rabies Immunization in Changsha, China: Results of a Cross-Sectional Cluster Survey, 2015–2021

**DOI:** 10.3390/v15010138

**Published:** 2022-12-31

**Authors:** Chunxiao Ji, Jia Feng, Siying Li, Hui Yang, Hui Wang, Xiangchang Geng, Hongliang Wang, Zengzai Liu, Tao Zhang, Yu He, Wei Liu

**Affiliations:** 1College of Veterinary Medicine, Hunan Agricultural University, Changsha 410128, China; 2Changsha Animal Disease Control Center, Changsha 410013, China; 3College of Bioscience and Biotechnology, Hunan Agricultural University, Changsha 410128, China

**Keywords:** rabies, dog, serum, vaccine, rabies antibodies, seroconversion, antibody level

## Abstract

The objective of this study was to examine longitudinal trends in the prevalence of dogs that are successfully immunized against rabies virus (as measured by sufficient serum antibodies) in Changsha, an urban center of China. The secondary objective was to investigate the factors affecting the seroprevalence of rabies virus antibodies in dogs. In this study, 4515 canine serum samples were collected from 57 pet hospitals (immunization points) during the period of 2015–2021 in five major urban areas of Kaifu, Furong, Tianxin, Yuhua, and Yuelu in Changsha, China. The enzyme-linked immunosorbent assay (ELISA) method was used to analyze the level and trend of rabies virus antibodies in serum and further evaluate the potential factors affecting the immunization effect from five factors: sex, age, time interval after most recent vaccination and sample collection, number of vaccinations, and vaccine manufacturer. The results showed that the seroconversion from the urban dog in Changsha steadily increased from 46.13% to 73.38% during 2015–2017. The seropositivity prevalence remained above the international standard (70%) from 2018 to 2020 and up to 90.99% in 2021. Further analysis showed that the seroconversion of rabies virus among dogs was significantly affected by the age, the number of vaccinations, time interval after the most recent vaccination and sample collection, and vaccine manufacturer, while sex had less influence. The overall rabies vaccination situation in urban areas of Changsha generally meets international standards, with only a few areas showing low levels of antibodies in dogs after vaccination and risk of infectiousness. Therefore, it is recommended that the first vaccination should be given when the dog is about three months old and regularly repeated every year after that. At the same time, antibody concentrations in dogs, especially in newborn puppies and older dogs, need to be tested promptly after vaccination at the required time to ensure that they are at a high level of immune protection, which can strengthen the supervision of rabies.

## 1. Introduction

Rabies, a highly lethal neurological disease caused by rabies virus (RABV) and rabies-associated virus [1,2,3,4], is one of the oldest zoonotic diseases in the world. Human rabies cases have been found in every continent except Antarctica, and 95% have occurred in Asia and Africa [5]. Rabies virus infection of the brain and spinal cord leads to 100% death once the disease develops. A total of 60,000 people die each year worldwide [6], so rabies control remains imminent. Led by the World Health Organization (WHO) in collaboration with the Food and Agriculture Organization of the United Nations (FAO), the World Organization for Animal Health (WOAH), and the Global Alliance for Rabies Control (GARC), the “Human Rabies Elimination” campaign aims to achieve zero human rabies death caused by dog rabies worldwide by 2030 [7,8]. Bites primarily cause the emergence and spread of rabies in humans and domesticated animals from rabid carnivores [1,9], and nearly 95% of human rabies infections are due to dog bites [10]. Therefore, strict surveillance of rabies virus infection in host animals, especially dogs, is imperative to reach the goal of zero rabies deaths by 2030 [11].

According to WHO, the rabies outbreaks have routinely been recorded in China, and the number of rabies cases in humans in different region of China is shown in Figure 1 (2014–2018), demonstrating that rabies is a constant public health threat in China [12,13,14]. According to the statistics, the number of rabies deaths in China peaked in 2007 with 3300 cases and declined to 290 cases in 2019 [1,15]. The main reasons for the decrease in mortality are the government’s management of post-exposure prophylaxis (PEP) and the increased level of rabies-related knowledge and awareness, which has led to more people making timely and relevant post-exposure management measures [12,16,17,18]. To reduce the incidence of rabies, in addition to prompt treatment after exposure, it is equally important to reduce the infection rate by the canine rabies virus, especially in pet dogs, which are most closely related to humans. The incidence of rabies in the individual provinces of China from 2014 to 2018 is shown in Figure 1, where the incidence rate ranges from low to high, color from light to dark [12]. The rabies epidemic is mainly concentrated in southern China, where Hunan Province continues to have a high incidence, with the highest number of cases in 2013 (83) and fluctuating lower thereafter, but with a slightly increasing trend in 2015 (75) and 2018 (78), and the following years with the number of cases in 2014 (65), 2016 (64), 2017 (71), 2019 (55) and 2020 (59), respectively [12], which is one of the most severely affected provinces in the current rabies epidemic.

Changsha, the capital city of Hunan Province, is in the middle of China, and mainly includes five urban areas, namely Kaifu, Furong, Yuelu, Tianxin, and Yuhua (Figure 1). According to the seventh census, the resident population of Changsha (urban areas) reached 3.482 million, and the area of Changsha (urban areas) is 434.82 square kilometers, and the population density is 8008 per square kilometer, according to gotohui statistics (https://www.gotohui.com, accessed on 20 November 2022). Moreover, the urban area of Changsha has good economic development with a significant GDP (USD 170 billion), it is an essential reference value for the metropolitan area survey of Changsha. According to the *Changsha Evening News*, by 2020, the number of pet dogs in Changsha reached 400,000, and the ratio of dogs to humans is 9:1. According to the Changsha City dog management regulations, you must apply to the public security authorities within 30 days from the date of purchase and inject a microchip and receive a dog ID card. From 2020, each urban area in Changsha has a dog detention center, the public can call 110 or report to the nearby police station when they encounter a stray dog, the public security organs are responsible for the capture and sent to the dog detention center, in addition to the public can also adopt these dogs. Under this policy, the number of stray dogs in the urban areas of Changsha has been greatly reduced. To prevent rabies, the Hunan Provincial Health and Wellness Commission recommends six measures for rabies PEP. (1) Immediately after being bitten, the wound should be repeatedly rinsed with 20% soapy water or 1% Neosporin (at that time, if there are no conditions, you can also use water to repeatedly rinse) for at least half an hour, trying to remove dog saliva and squeeze out the dirty blood, but never use your mouth to suck the dirty blood from the wound. (2) After rinsing, use medical alcohol or iodine. (3) When dealing with the wound inside the eye, rinse with sterile saline and generally do not use disinfectant. (4) Treatment of oral wounds should be performed with the assistance of an oral surgeon. (5) If the wound is large and deep, with more bleeding, go to the hospital immediately. (6) After going to the hospital, the doctor will decide whether rabies vaccination is needed. The doctor will decide if rabies vaccine and immunoglobulin or immune serum are needed.

Vaccine immunization is the most effective approach for preventing and controlling canine rabies, and the level of antibody after immunization is an important indicator for evaluating the effectiveness of vaccine immunization [19,20]. According to the Changsha City Dog Breeding Management Regulations, puppies are required to have their initial rabies vaccination at three months, a second vaccination at one year of age, and subsequent vaccinations every other year, and regular testing of antibody levels in dogs to ensure a positive serum antibody rate. In this study, 4515 sera were collected to investigate whether rabies serum antibody levels can effectively resist rabies and the trend of seropositivity in dogs in Changsha (2015–2021). In addition, we conducted statistical analyses to understand correlations between assay results and characteristics of the dogs providing the sera, in order to provide additional recommendations for canine rabies prevention and vaccination in Changsha.

## 2. Materials and Methods

### 2.1. Sample Recruitment

A total of 4515 samples were collected from 57 pet hospitals in all 5 urban areas of Changsha (2015–2021). We selected 57 hospitals cooperating with Changsha Animal and Plant Disease Prevention and Control Center as immunization sites, which are strong and have many classic cases and sample data, and the sample size collected from these large veterinary hospitals is sufficient to represent other veterinary hospitals in Changsha. The selection criteria of the selected veterinary hospitals were (1) covering all areas, (2) having the qualification of vaccination approved by local authorities, (3) different regional scope, (4) good conditions, etc., to ensure the accuracy of the samples and the representativeness of the study results. The hospitals serving as immunization sites also administered rabies vaccine to dogs and regularly monitored serum antibody levels. As the number of pet hospitals increases gradually, the number of immunization sites (pet hospitals) cooperating with the center increases year by year. The specific information is shown in Table 1.

Most of the 4515 serum samples in this survey were obtained from routine health visits to pet hospitals (immunization sites), and there were also stray dogs captured by government agencies and sent to dog holding facilities, so there might be cases where multiple samples came from the same dog. In addition, when serum samples were tested for serological antibodies, the samples were recorded for sex, age, time interval after most recent vaccination and sample collection, number of vaccinations, and vaccine manufacturer.

### 2.2. Seropositivity Test and Analysis of Antibody Titres

This study used ELISA kits (Beijing Genomcell.bio. Hangzhou, China) for quantitative serum antibody testing. A solid-phase enzyme immunoassay method known as an indirect ELISA is the foundation of the ELISA test kit. Rabies glycoprotein, which was taken from was purified from G protein expressed in vitro, was applied to a microplate. The enzymatic conjugate comprised peroxidase and protein A from Staphylococcus aurius. The specific steps were performed in standard 96-well microplates coated with RV glycoprotein according to the manufacturer’s instructions. After adding the blocking solution, the optical density was measured at 450–620 nm with an ELISA reader 30 minutes later, and the criteria for determination are shown in Table 2. According to the WOAH standard, dogs are considered to have the ability to resist rabies virus when their serum antibody titers are ≥ 0.5 IU/mL. Since this investigation used the ELISA indirect method, which is not a complete response to the rabies antibody titer level, according to the instructions, we considered Y ≥ 0.5 EU/mL to be resistant to rabies.

### 2.3. Statistical Analysis

We obtained a total of 4515 serum antibody level test results and related information (age, time interval after most recent vaccination and sample collection, number of vaccinations, and vaccine manufacturer). This information is obtained by the owner or the person in charge filling out a questionnaire, which is required for dog filing, health visits or dog illness, but the information was lost due to the lack of detailed recording of sample information at some immunization sites or the long-time span, which eventually led to different numbers of samples corresponding to different characteristic (Table 3). The data were entered into the database using EpiData Entry v.3.1 (EpiData, Odense, Denmark), and analyzed descriptively by SPSS 20.0. and the Pearson chi-square test was used to determine the significance and relatedness of factors associated with seropositivity. Statistical significance between categorical variables was tested at the 5% significance level (*p* < 0.05).

## 3. Results

### 3.1. Immunization Sites and Number of Samples

During the seven years (2015–2021), 4515 samples were obtained from 57 sampling sites in five urban areas of Changsha. Overall, the number of immunization sites increased from 17 (2015) to 57 (2021), and the number of serum samples also showed an increasing trend between 2015 (336) and 2021 (955). The trends are shown in Figure 2 and the quantity is shown in Table 1. As mentioned earlier, the number of samples corresponding to different Characteristic varied due to the loss of related information caused by the lack of detailed recording of serum sample information at some immunization sites or the long-time span, and the specific number shown in Table 3.

### 3.2. Rabies Seroconversion

During the period 2015–2021, the proportion seropositive to rabies in the five urban areas of Changsha rose from 46.13% in 2015 to more than 50% in 2016 and dynamically increased during this period, remaining above 70% from 2017 to 2020 and reaching an ultra-high seropositive proportion of 90.99% in 2021. Overall, the seroconversion in each district dynamically increased from 2015 to 2021. Among them, Tianxin and Yuelu fluctuated more severely from 2015 to 2018, fluctuating between 45% and 65%, with a slight upward trend through regression analysis. While Kaifu, Furong, and Yuhua in 2015–2018 displayed an apparent upward trend, in 2017 and 2018 its seroconversion was maintained above international standards, and even in 2018, it all rose to more than 80%, far exceeding international standards. However, in 2019 the qualified rate of rabies antibodies in these three districts had different degrees of decline. Meanwhile, the seroconversion of Tianxin increased and stabilized at more than 70% in 2020 and 2021 (Figure 3). The specific quantities are shown in Appendix A.

### 3.3. Factors Affecting Rabies Seroconversion 

A total of 4515 serum samples were obtained in this survey. Serum-related information (sex, age, number of vaccines, etc.) was collected in varying amounts (Table 4).

#### 3.3.1. Sex

The results of antibody testing of these 1382 serum samples were categorized by sex, of which 788 samples were male, and 571 samples tested positive (72.46%). There were 594 female dogs, and 416 samples tested positive (70.03%) (Table 4). A t-test was performed between the two groups, and statistics revealed that the dog ‘s sex was not associated with seroconversion (Figure 4).

#### 3.3.2. Age

There were 784 serum samples of the known age of the dogs, and a comparative analysis of the test results revealed that the rabies seroconversion was significantly correlated with age (Table 4). The seroconversion for dogs under 6 months of age was 62.86%, 60.11% for dogs at the 6 to 12 months of age, 71.60% for dogs at the 1 to 2 years of age, 72.38% for dogs at the 2 to 5 years of age, and 79.78% for dogs over 5 years of age. The results showed that the age of the dog was associated with the rate of rabies seropositivity (*p* = 0.006**, < 0.05) (Figure 4). 

#### 3.3.3. Number of Vaccinations 

A total of 1380 serums were tested for seroconversion and sorted according to the number of vaccinations. The results revealed that the lowest seroconversion was observed for 1 vaccination (61.43%) and the highest for 4–6 vaccinations (83.33%). The rate decreased to 73.91% for more than 7 times of repeated vaccination (Table 4). The Chi-square test revealed a statistically significant difference (*p* = 0.00001***, <0.0001) in the rabies seroconversion for different numbers of vaccinations (Figure 4).

#### 3.3.4. Time Interval after Most Recent Vaccination and Sample Collection 

A total of 1182 serum samples with definite vaccination time and sampling time were statistically analyzed and classified into three categories according to the sampling time after vaccination, within 1 month, 2–6 months and 7–12 months. The test results showed that the seroconversion was 60.67% (182/300), 67.97% (399/587), and 59.66% (176/295), respectively (Table 4). The results showed that the time interval after the most recent vaccination and sample collection was associated with the seropositivity rate (*p* = 0.0194 *, < 0.05) (Figure 4).

#### 3.3.5. Vaccine Manufacturers

A total of 1369 serum samples were categorized according to vaccine manufacturers to study the effect of different brands of vaccines on seropositivity prevalence. The seroconversion of the French B vaccine was the highest (77.45%), and the seroconversion of both the French B and China A vaccines was above 70%, reaching the international standard. The Dutch A vaccine observed the lowest seropositivity (65.18%) (Table 4). The chi-square test revealed a statistically significant difference (*p* = 0.005 **, < 0.05) in the seroconversion of different vaccine manufacturers (Figure 4).

## 4. Discussion 

According to the World Health Organization, strengthening immunization against rabies is a fundamental measure to prevent rabies, and the effective prevention and control of rabies can be achieved if the immunization coverage rate is above 70% [21,22,23,24]. To investigate the effectiveness of rabies immunization in pet dogs in urban areas of Changsha, a total of 4515 dog serum samples were tested during 2015–2021. With the development of society and the increase in the number of pet owners, the test sites have increased significantly, accompanied by an increase in sample size. During the seven years, although the immunization effect in 2015 and 2016 was poor and the seroconversion did not reach the standard of 70%, the seroconversion was maintained at about 70% after 2017 and reached 90.23% in 2021. The results indicate that the vaccination of urban pet dogs has gradually matured, the prevalence and efficiency of immunization have been significantly improved, and an effective defensive barrier has been initially established. The increase in rabies seropositivity is mainly due to the enactment of relevant government policies and greater public awareness of rabies. The scope of this survey was five urban areas of Changsha, which was only limited to the metropolitan area of Changsha. The results could only reflect the rabies antibody level of pet dogs in the urban area but not the whole of Changsha. However, since Changsha is one of the new first-tier cities in China and is strategically located, this study has significant reference value for urban rabies control in other cities in China and even in developing countries where dog rabies exists as India and Brazil.

Another aim of this investigation was to derive the factors influencing the seropositivity prevalence through statistical analysis of five aspects: sex, age, number of vaccinations, time interval after most recent vaccination, sample collection, and vaccine manufacturer. The findings showed that sex has no significant correlation with the rate of seropositivity in pet dogs, which is similar to the results of Bhumika F Savaliya [24] and Louise T Berndtsson, et al. [25]. While age, the number of vaccines, time since vaccination, and vaccine manufacturer have significant correlation. In addition, one of the limitations of this survey is that we were unable to compare the seropositivity rates of stray and domestic dogs because we were unable to determine whether the sample information came from stray or domestic dogs due to the long period of time.

The two trends coincided from the results of antibody testing at different ages and the number of vaccinations. The rabies proportion seropositive of dogs increased with the age of the dogs and the number of vaccinations. Armin Saalmüller and G Sage et al. found that dogs vaccinated only once may fail to immunize or produce low antibody levels, which is consistent with our results [26,27]. In addition, our data showed that 73.31% of the pet dogs that were immunized once were under one year old, and we consider that the reason for the lower protective anti-rabies antibody levels was caused by the combination of age and the number of vaccinations; dogs immunized two times or more had significantly higher antibody levels. Dogs vaccinated four to six times had the highest seroconversion (83.33%), indicating that the higher the number of vaccinations, the higher the seroconversion. P Ntampaka et al. conducted a cross-sectional survey of 137 dogs and showed that booster vaccination significantly increased antibody levels [28]. In our survey, the seroconversion of the sample with ≥ 7 immunizations decreased to 73.91%, which was thought to be due to a small sample size of less than seventy, or due to the age and physical decline of the dogs with 7 immunizations, and this result was too high by chance to be statistically significant. Ippei Watanabe et al. showed that 51.1% of dogs receiving 1 rabies vaccination had protective virus neutralizing antibody (VNA) levels (≥ 0.5 IU/mL) with a geometric mean of 0.61 IU/mL [29]. In contrast, 97.8% of dogs vaccinated at least twice had protective VNA levels with a geometric mean of 7.86 IU/mL. In addition, 97.9–100% of dogs retained protective VNA levels at least twice in the second year after the last vaccination. Although VNA levels in dogs vaccinated at least twice declined gradually over the 2 years following the previous vaccination, 78.9% of dogs retained protective VNA levels [29,30]. Therefore, it is recommended that pet dogs should be vaccinated regularly every year to maintain rabies antibody concentrations in pet dogs.

Our study showed that the rabies seroconversion in pet dogs over six months increased with age, while the seroconversion in pets aged less than six months was greater than that in pets aged 6–12 months. According to our analysis, pet dogs are born with protection from maternal antibodies in their bodies, and this protection is more pronounced in pet dogs up to 6 months of age. However, the concentration of maternal antibodies decreases with age, so their seroconversion is slightly higher than that of pet dogs aged 7–12 months [31,32]. The statistical analysis of older dogs older than seven years was not performed because the sample size of older dogs was too small to be statistically significant, so it was impossible to determine whether the seroconversion would decrease as immunity and all functions of the body decreased due to continued aging. Louise T Berndtsson et al. studied 6789 rabies-vaccinated dogs and showed that seropositivity prevalence were lower in dogs under one year and over five years of age [25]. However, according to other literature, in the case of older dogs, the decrease in serum rabies antibodies per unit of time will accelerate with age, all else being equal [32,33,34]. Therefore, it is recommended that dogs be vaccinated after 3–6 months of age when they are more likely to produce high levels of antibodies.

In addition, the seropositivity rate increased and then decreased with the time interval after the last vaccination and sample collection. This was due to a gradual increase in serum antibody titers after the last vaccination and then a gradual decrease over time, but we were unable to obtain a similar conclusion in our current investigation. This is because the ELISA method was used in this serum antibody titer survey, which does not accurately measure antibody titers and only measures the IgG response to the anti-rabies glycoprotein, which occurs after the IgM response, and both IgM and IgG antibodies can neutralize the virus. We could only conclude that the serum antibody positivity rate was related to the time interval, and by analyzing the correlation between the time interval and the serum antibody titer, we laterally inferred that the serum antibody positivity rate was due to a decrease in the serum antibody titer. The results of this investigation showed a higher rate of seropositivity in dogs with a time interval of 2–6 months between vaccination and testing than in those with a time interval of less than 1 month, because Elisa only detects IgG antibodies in the serum. Many studies have shown that the body produces IgM antibodies about 7 days after rabies vaccination [35], IgG after about 14 days [35,36], and reaches a maximum between days 30 and 40 [37], after which it gradually decreases. Rabies virus IgM/IgG conversion was observed after day 10 [38]. In addition to this the results showed that the seropositivity rate was less than 70% for all Elisa performed within 12 months of vaccination. This may be since many of the 1182 samples provided by the pet hospitals were from dogs that were first vaccinated against rabies and most of the serum samples were from before 2019. However, the exact reason for this requires further study. It is worth mentioning that RFFIT and FAVN are commonly used internationally to detect serum rabies antibody titers, both of which can detect both IgM and IgG neutralizing antibodies in the serum. Lorna J Kennedy et al. tested 10,438 serum samples for antibody titers by fluorescent antibody virus neutralization (FAVN). It was found that as the time from vaccination to sampling increased, more and more dogs failed to reach the antibody response threshold [39]. In addition to this, R.M.S. Pimburage et al. investigated the rate of seropositivity to canine rabies virus neutralizing antibodies after vaccination by the Rapid Fluorescent Focus Inhibition Test (RFFIT) method. The results showed that serum antibody titers reached their highest at 30 days after vaccination and gradually decreased thereafter [28]. This study further showed that serum antibody titers increased and then diminished over time. Serum antibody titers were low for two weeks after vaccination, and only after 21 d of immunization the body will gradually produce a higher concentration of antibody, while antibody titer was highest after 2–6 months of vaccination and gradually decreased after 7 months, which is in line with the law of reducing immunization antibody [40,41,42]. Furthermore, many studies have shown that a single subcutaneous injection of tissue culture vaccine in dogs does not guarantee that serum antibodies will meet international standards for the next year [28,30,43]. Therefore, a booster immunization is recommended 1 month after the initial vaccination and again annually after that.

This study showed significant differences in seropositivity prevalence for vaccines from different manufacturers. However, the seroconversion of most manufacturers’ vaccines was above 70%, and only some manufacturers’ vaccines had poor immunization effects. According to the analysis, this is because the number of immunization samples from some manufacturers is too small, which may impact the test results. It is also reported in the relevant literature that the immunization effect of each brand of Vaccine is comparable [44,45], even though there is a slight difference in the price of vaccines in different countries and regions. Still, it does not have a significant impact on the seroconversion. Therefore, the public can choose according to their actual situation.

## 5. Conclusions

The results showed that in the past 7 years (2015–2021), the seroconversion proportion of pet dogs in urban areas of Changsha City has increased yearly and reached international standards. Serum antibody levels were associated with vaccine age, a number of vaccinations, manufacturer, time interval after most recent vaccination, and sample collection, but not with sex. Based on these findings we recommend the government needs to insist on rabies vaccine monitoring in Changsha to further increase the rate of positive dog serum antibodies to ensure the completion of the rabies 2030 clearance program. For pet owners, they should adhere to the annual rabies vaccination schedule and conduct timely antibody testing to immunize their dogs. 

## Figures and Tables

**Figure 1 viruses-15-00138-f001:**
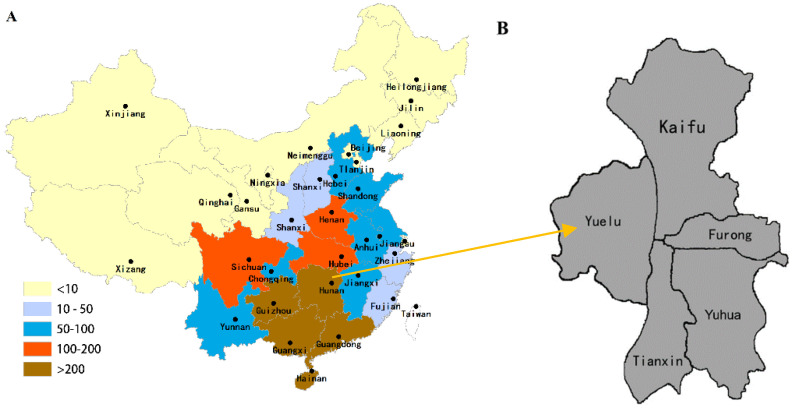
The regional distribution of averaged number (year) of rabies cases (**A**) (the figure is from Yong-Chao Qiao, et al. [12]) and administrative division, Changsha urban area, China, 2021 (**B**).

**Figure 2 viruses-15-00138-f002:**
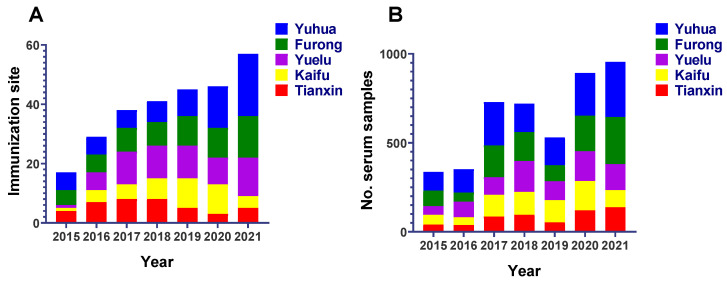
The number of immunization sites (pet hospitals) involved in the study (**A**) and the number of dog sea collected for rabies antibody testing (**B**) in Changsha, China, 2015–2021.

**Figure 3 viruses-15-00138-f003:**
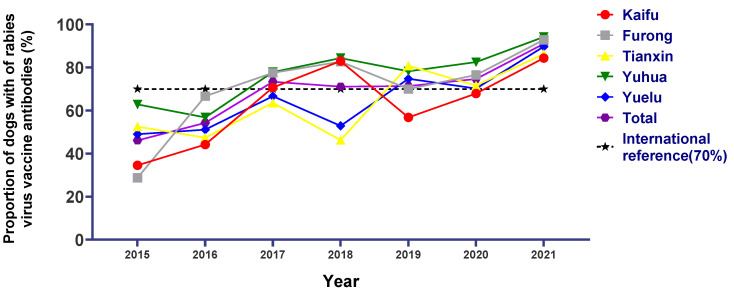
Proportions of seroconversion to vaccination against rabies virus over time in each urban area of Changsha, China, 2015–2021.

**Figure 4 viruses-15-00138-f004:**
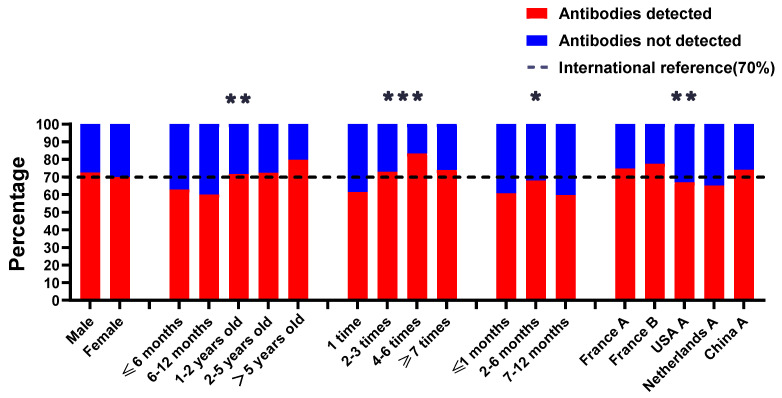
Effect of different factors on the qualification rate of rabies vaccine. Note: * Significant differences exist between variable categories at 5% significant level. ** Significant differences exist between variable categories at 1% significant level. *** Significant differences exist between variable categories at 0.1% significant level.

**Table 1 viruses-15-00138-t001:** Immunization points (No. of sites) and sample size (No. of dogs), Changsha, China, 2015–2021.

	2015		2016		2017		2018		2019		2020		2021	
	No. of Sites	No. of Dogs	No. of Sites	No. of Dogs	No. of Sites	No. of Dogs	No. of Sites	No. of Dogs	No. of Sites	No. of Dogs	No. of Sites	No. of Dogs	No. of Sites	No. of Dogs
**Kaifu**	1	55	4	43	5	123	7	129	10	125	10	165	4	96
**Furong**	5	87	6	51	8	178	8	162	10	90	10	200	14	265
**Tianxin**	4	40	7	38	8	85	8	95	5	52	3	120	5	138
**Yuhua**	6	105	6	132	6	244	7	160	9	156	14	240	21	310
**Yuelu**	1	49	6	88	11	99	11	174	11	107	9	168	13	146
**Total**	17	336	29	352	38	729	41	720	45	530	46	893	57	955

Note: No. of sites: Immunization sites; no. of dogs: number of samples.

**Table 2 viruses-15-00138-t002:** Determination criteria for quantitative testing.

	Positive	Weak Positive	Negative
**Quantitative Test**	Y ≥ 0.5 EU/mL	0.125 EU/mL ≤ Y < 0.5 EU/mL	Y < 0.125 EU/mL

Note: Y: Serum antibody level of the sample to be tested.

**Table 3 viruses-15-00138-t003:** The different background information of the serum samples corresponds to the number of samples.

Characteristics	Values	Numbers Sampled
**Sex**	Male	788
	Female	594
**Age**	≤6 months	35
	6–12 months	183
	1–2 years old	162
	2–5 years old	315
	≥5 years old	89
**Number of Vaccinations**	One time	490
	2–3 times	551
	4–6 times	270
	≥7 times	69
**Time Interval after Most Recent Vaccination and Sample Collection**	≤1 month	300
	2–6 months	587
	7–12 months	295
**Vaccine Manufacturer**	France A	593
	France B	102
	USA A	184
	Dutch A	382
	China A	108

**Table 4 viruses-15-00138-t004:** Potential factors associated with seroconversion in dogs.

Characteristic	Value	Number with Detectable Vaccine Antibodies	Number Sampled	Proportion Seropositive	X^2^	Significance (*p*-Value)
**Sex**	Male	571	788	72.46%	0.978	**0.323 ^NS^**
	Female	416	594	70.03%		
**Age**	≤6 months	22	35	62.86%	14.392	**0.006 ^**^**
	6–12 months	110	183	60.11%		
	1–2 years old	116	162	71.60%		
	2–5 years old	228	315	72.38%		
	≥5 years old	71	89	79.78%		
**Number of Vaccinations**	One time	301	490	61.43%	43.006	**0.00001 ^***^**
	2–3 times	402	551	72.96%		
	4–6 times	225	270	83.33%		
	≥7 times	51	69	73.91%		
**Time Interval after Most Recent Vaccination and Sample Collection**	≤1 month	182	300	60.67%	7.882	**0.0194 ^*^**
	2–6 months	399	587	67.97%		
	7–12 months	176	295	59.66%		
**Vaccine Manufacturer**	France A	444	593	74.87%	14.731	**0.005 ^**^**
	France B	79	102	77.45%		
	USA A	123	184	66.85%		
	Dutch A	249	382	65.18%		
	China A	80	108	74.07%		

Note: ^NS^ = Non-significant. * Significant differences exist between variable categories at 5% significant level. ** Significant differences exist between variable categories at 1% significant level. *** Significant differences exist between variable categories at 0.1% significant level.

## Data Availability

The data presented in this study are deposited in the Changsha Animal Disease Control Center, Changsha, Hunan 410013, China and available on request from the corresponding author.

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
