# Peer review of "Factors Associated with Dog Rabies Immunization in Changsha, China: Results of a Cross-Sectional Cluster Survey, 2015–2021"

_viruses, 2022, doi:10.3390/v15010138_

Round 1
Reviewer 1 Report
Ji and colleagues investigate the prevalence of dogs with sufficiently high titers of rabies vaccine antibodies in Changsa, China as well as various factors (e.g., sex, time since vaccination, dog age) associated with this prevalence. The strengths of this study are the large sample size (>4,500 dogs) and the longitudinal nature of the study (2015-2021). Changsa is a relevant location, as it is a highly populous location where incidence of rabies in humans is high. Dog vaccination is a key strategy for the elimination of rabies; thus the topic is of importance.
However, the study is omitting key details about the sampling sites and the dogs themselves. Without these details, it is difficult to examine how generalizable these dogs are to the wider dog community in Changsa. It is also not clear if all dogs in the study have been vaccinated against rabies. This makes it hard to fully interpret the objective and implications of the study. Finally, a multivariate model would be more appropriate to examine factors associated with seropositivity. My major recommendations are below, followed by minor comments.
Major comments:
In the introduction, some more information about Changsa may be interesting. For instance:
- what is the population size? The geographic area?
- Is there any data on the ratio of dogs to humans in Changsa, or availability of PEP in Changsa?
- How common are free-roaming or stray dogs?
- Is rabies vaccination of pet dogs mandatory in China? If so, how frequently?
In Methods section 2.1, the authors should include more information about the pet hospitals chosen:
- Over what years were samples collected, and how many samples per year? I see this information in Table S1, but it would be helpful to have in the main text. It would be helpful as well to reference Figure 2 in this section.
- How many pet hospitals were chosen?
- Wy does the number of sites change over time? Do some hospitals drop out and re-enter the study? More on the design of site selection and whether sites left and re-entered the study is needed.
- It indicates that the hospitals were chosen for their excellent infrastructure. Are these pet hospitals representative of other pet hospitals?
- The phrase “pet hospital” is used in the text, while “immunization sites” is used in figure legends. The authors should be consistent with which one is more accurate. If pet hospitals also function as immunization sites, the authors should indicate so, while still using the phrase “pet hospital”
In Methods section 2.1, the authors should include more information about the dogs within the pet hospitals:
- How were dogs selected from the pet hospitals?
- More information about dogs that are brought to the hospitals is needed. For instance, are only very sick dogs brought to the hospital, or are dogs brought for routine health visits?
- I assume only owned dogs are brought to the hospitals – is this correct?
- Were any dogs sampled more than once over the sampling period?
Additional factors were only obtained on a small subset of dogs. Can the authors explain in the methods how they obtained additional information on the dogs and why the information is available for only a subset? (perhaps Methods section 2.3 is where this information could be added)
While the univariate comparisons are interesting, a multivariate logistic regression model would be stronger, as individual associations may be confounded by the other factors considered. For instance, it is possible that certain hospitals prefer certain vaccine manufacturers, and thus the effect of vaccine manufacturer is confounded by hospital location. In running the multivariate regression, the authors should ensure to properly adjust for year and clustering by pet hospital in the model. While the author could still include univariate comparisons, the results from an adjusted model would be more informative than the results displayed in sections 3.3.
Are all dogs in the study vaccinated against rabies? I initially read the study thinking that not all dogs were vaccinated, but the heading of Table 2 implies that they are. Understanding this point dramatically changes the comparison of these results to the 70% immunization coverage. If the dogs are all vaccinated, then comparison of the 90% seroprevalence to the 70% recommended immunization coverage may not be appropriate. Moreover, if all dogs are vaccinated, the objective of the study should be described as examining factors associated with sufficient antibody titers among vaccinated dogs.
The discussion must consider the generalizability of results, particularly with respect to the population sampled and limitations discussed, where appropriate. People who bring their dogs to pet hospitals may not represent the general pet owning population, for instance.
Minor comments:
Abstract
In the abstract (line 17), the authors should include information about where the sera were collected (e.g., replace “57 sites” with “57 pet hospitals”.
Line 16: Please add that the study concerns “factors affecting the seroprevalence of rabies virus antibodies” in dogs (or factors affecting the prevalence of sufficient antibody titers in vaccinated dog)
Introduction
The sentence line 59 is a bit confusing me to: “To reduce the incidence of rabies, in addition to prompt treatment after exposure, it is more important for reducing the infection rate by the canine rabies virus, especially in pet dogs, which are most closely related to humans”. Are the authors implying that reducing infection rate in canines is more important than PEP? The authors should rephrase to make their point clear.
The information about the incidence rate of rabies in humans over time is interesting. The authors might consider adding a figure of the incidence rate of rabies over time in Changsa (or Hunan, if that data is not available), either in the main text or in the supplement.
Methods
Please see major comments above, most of which pertain to methods.
In Section 2.2, please describe whether you consider weak positives as positives, or only positives.
Results
Can the authors clarify this sentence, line 108: “T-test was used to detect the effect of gender of vaccine production on the seropositivity rate”
Line 130 says, “Among them, Tianxin and Yuelu fluctuated more severely from 2015-2018, fluctuating between 45% and 65% without a significant upward trend.” From Figure 4, I do see an upward trend. The authors may wish to formally test the trends over time using regression methods.
Information about the dogs shown in Table 2 should be presented earlier in the results. For instance, Section 3.1 could include information about the sites, the number of samples, and the characteristics of the dogs sampled. If the authors could include information about why the dog was attending the hospital, that would be appreciated as well.
In all 3.3 sections, the authors use language that implies causation, such as: “gender had no significant impact” or “results showed a significant effect of age on rabies seroprevalence”. Causality cannot be inferred from this study. The authors should please rephrase this as an association. (e.g., “the sex of the dog was not associated with prevalence of antibodies”)
Discussion
Line 194 states: “the effective prevention and control of rabies can be achieved if the immunization coverage rate is above 70%” Can the authors clarify if this is 70% for all dogs – including free-roaming dogs. Are free-roaming dogs a challenge in this population?
It is encouraging to see seroprevalence >90%. However, I am confused if all dogs included in this study are vaccinated. If so, this number cannot be compared to the immunization coverage rate of 70%.
Line 199: the number 4,377 does not match the rest of the manuscript
The increasing trend in antibody titers over time is encouraging. Where there any interventions or campaigns in this area that the authors think may have contributed to this increase?
Throughout the manuscript:
Since it is assumed that the differences in gender are comparisons between the biological factor, sex, all instances of “gender” should be changed to “sex”.
Instances of “rate” in the manuscript are more accurately described as “prevalence” or “proportions”. For instance, “Seropositivity rate” or “seroprevalence rate” or “qualification rate” would be more accurately represented as a “seropositivity prevalence” or “the proportion seropositive”, or simply, “seroprevalence”. This phrase is used multiple times throughout the manuscript. If the study is only among vaccinated dogs, I recommend, “prevalence of sufficient antibody titers”, or something similar.
Figure 1.
In the pdf, it is not possible to clearly see the provinces written. The authors should ensure the names are legible in their version.
Figures 2 and 3 are repetitive. I suggest to keep Figure 2 and describe totals in the text. I also suggest to change the caption to: “Figure 2. The number of pet hospitals involved in the study (A) and the number of dog sea collected for rabies antibody testing (B) in Changsha, China, 2015-2021.”
Figure 4. I suggest to change the y axis title to “Proportion of dogs with of rabies virus vaccine antibodies (%)”, and to change the legend to, “Prevalence of rabies virus vaccine antibodies over time in each urban area of Changsha, China, 2015-2021”
Table 1 column headings would make more sense as “Characteristic”, “Value”, “Number with detectable vaccine antibodies”, “Number sampled”, “proportion seropositive”, “2”, “p-value”
Figure 5. I recommend to change the legend to “univariate associations between different factors and detection of rabies vaccine antibodies in dogs” and change the legend to “antibodies detected” and “antibodies not detected”

Author Response
Thank you very much for your favorable comments and constructive suggestions. These comments and suggestions are very valuable for us to revise and improve the quality of our MS. We have revised and improved the MS strictly according to the reviewers’ comments and suggestions.
Please find the attachment for details.

Reviewer 2 Report
The paper titled “Factors associated with dog rabies immunization in Changsha, China: results of a cross-sectional cluster survey, 2015-2021”, though similar at first glance to other papers on seroprevalence data in a dog population presents survey in an area not previously investigated. The survey was completed to determine if the efforts affecting rabies vaccination in dog population for rabies control were successful.
The study presented is straightforward but there are some areas that require correction and improvement.
General, overall comments:
The test method used for serology is an indirect ELISA, however the results are reported in IU/mL instead of EU/mL. IU/mL is referencing the international WOAH (formerly known as of OIE) reference serum, the potency of this reference was determined by FAVN, a serum neutralization (SN) method. ELISA and SN methods do not measure the same function of rabies antibodies; thus, Equivalent Unit is the correct unit when referring to ELISA results. In addition, ELISA results are not referred to as titers, throughout the paper results are referenced as such. This makes it confusing for the reader to understand what results are being mentioned.
The text often refers to the results meeting the ‘international standards’. I believe what is meant is the 0.5 IU/mL rabies virus neutralizing antibody level. This it the level recognized by WHO and WOAH as the representing adequate response to rabies vaccination. It should be clearly stated in the methods that the serology results represent EU/mL and the authors are using 0.5 EU/mL as equivalent to IU/mL (see line 101). Further needed is explanation on how the equivalency was determined. The reference given for the method (21) refers to the FAVN method not an ELISA. Either the method section for the serology assay needs to be fully described or a valid reference used.
The use of protection levels or protective level is not accurate, there is not international level for rabies protection, only level representing adequate vaccination response, please correct throughout the manuscript.
Specific comments:
Line 45, to be up to date with the current abbreviation, change OIE to WOAH.
Line 46, change the sentence to “….the ‘Human Rabies Elimination” campaign aims to achieve zero human rabies death worldwide by 2030.” to “….the ‘Human Rabies Elimination” campaign aims to achieve zero human rabies death caused by dog rabies worldwide by 2030.”
Line 63 add a comma after high for clarity
Line 66, suggest changing ‘severe’ to ‘severely affected’
Line 68-69, the sentence is unclear, suggest rephrasing.
Line 77-78, incomplete sentence.
Line 86, what is meant by high accuracy?
Table 1 states that the criteria are quantitative, but the levels given are qualitative? It would be helpful to have both qualitative and quantitative (linear range, etc.) criteria listed.
Line 88, the vaccine is referred to as Dutch here but in the Table 2 it is listed as Netherlands, consistency is preferred.
Lines 260-276, seemingly there is no recognition that comparing SN (FAVN or RFFIT) results to ELISA results is not valid especially for an indirect ELISA measuring only the IgG response. Also, it is unclear why references 38 and 39 were referenced as 38 is referring to human study using a monoclonal and 39 is referencing a study of dog bite incidence and no serology results were included.
Line 270, similar to the above, it is not clear why reference 40 is referred to here as it is a study in humans and did not measure rabies antibodies at day 21.
Author Response
Thank you very much for your favorable comments and constructive suggestions on our manuscript (MS) viruses-2014760. We have revised and improved the MS strictly according to the reviewers’ comments and suggestions. Please find the attachment for details.

Round 2
Reviewer 1 Report
The clarity of the paper is much improved. However, there remain several areas where I believe clarity can be improved. In the attached document, I detail some of those locations.
Additionally, I am surprised at how low seroconversion of recently vaccinated dogs are, given that >90% seroprevalence was seen in recent years. Can the authors verify this finding has been correctly analyzed?

Author Response
Thank you for your comments, please see the attachment for details.

Reviewer 2 Report
Thank yuou for the corrections/edits which have improved the paper. I have one suggestion to improve the understanding of the serology results. In lines 337-340, it would be helpful to note that because the ELISA used measures only the IgG response of anti-rabies glycoprotein which occurs after the IgM response, this explains why the results demonstrate a low antibody level until day 21, which is different from studies that use SN methods such as RFFIT and FAVN, which measure both IgM and IgG neutralizing antibodies.
Author Response
General, overall comments:
POINT 1 In lines 337-340, it would be helpful to note that because the ELISA used measures only the IgG response of anti-rabies glycoprotein which occurs after the IgM response, this explains why the results demonstrate a low antibody level until day 21, which is different from studies that use SN methods such as RFFIT and FAVN, which measure both IgM and IgG neutralizing antibodies.
Responses: Thanks to your suggestion, we have added relevant content to the article to make it more logically coherent and clear. Please see the article for details(Lines 323-361). First, we clarified that this investigation was conducted using the Elisa method, which can only detect IGg in serum. In addition to this, we compared FAVN and RFFIT (which can detect IgG and IgM in serum) to complement the pattern of antibody production after vaccination and concluded that regular repeat vaccination is needed.
Thank you for your advice, which is very helpful to me.